# Management Approaches to Health and Safety at Work during Prevention Intervention Planning

**DOI:** 10.3390/ijerph20247142

**Published:** 2023-12-05

**Authors:** Vanessa Manni, Diego De Merich, Giuseppe Campo

**Affiliations:** Department of Medicine, Epidemiology, Occupational and Environmental Hygiene, Inail, 00143 Rome, Italy; d.demerich@inail.it (D.D.M.); g.campo@inail.it (G.C.)

**Keywords:** organizational models, prevention intervention, efficacy evaluation, (worker) participation

## Abstract

This work deals with a systematic review of the literature data concerning the theme of integrated approaches to occupational health and safety management, with particular reference to the programming of assistance plans, which guide companies’ organizational choices by also addressing the principles of Total Worker Health. In the current regulatory framework on this issue, the UNI ISO 45001: 2018 standard “Occupational health and safety management systems—Requirements and guidance for use” (published on 12 March 2018)” assumes relevance, defining dynamic approaches to occupational health and safety management systems—paying particular attention to external contextual factors that may influence corporate organizational decisions. The adoption of these systems is not mandatory but allows companies to fulfill their duties in terms of health and safety at work through an organizational approach aimed at the awareness, involvement, and participation of all subjects of the company prevention system, progressing past the phase of mere technological and prescriptive approaches towards a holistic vision of prevention that places the person at the center of preventive actions. In this context, the role of institutional networks and socio-economic partnerships assumes particular importance via the activation of territorial assistance interventions to support companies aimed at improving risk management levels. To this end, the importance of verifying the effectiveness of assistance interventions emerges from the scientific debate using indicators such as quantitative indicators aimed at measuring the performance of all phases of an intervention, with particular attention to their effects in terms of the improved solutions developed.

## 1. Introduction

### 1.1. The International, European and National Framework

The approval of the UN Agenda 2030 represented an evolution towards a combined approach in which all objectives consider economic, social, and environmental aspects and aim to put an end to poverty, restore dignity to people and, at the same time, preserve nature and the environment. The Ostrava Declaration highlights the need to strengthen commitment at international and national levels to improve environmental protection strategies and to prevent the adverse effects, costs and inequalities of conditions that impact the environment and health.

The European Commission, through the “Strategic Framework on Health and Safety at Work 2021–2027” published on 28 June 2022, has defined the priorities and key actions needed to improve the health and safety of workers, following rapid changes in economy, demographic evolution, and work models.

This new strategy focuses attention on three transversal objectives, highlighting—in the introduction—how the protection of workers’ health and safety is above all a right, but also represents a solution to respond to the new challenges posed by transformations of the world of work, from digitalization to demographic changes.

Among the objectives is that of improving the prevention of accidents and occupational diseases, introducing—in the European panorama—the concept of improvement already launched in previous strategies and implemented by the Italian legislation on the subject: Legislative Decree 81 of 2008.

On the Italian front, the Ministry of Health with the National Prevention Plan (PNP) 2020–2025 (Adopted with the State-Regions Agreement of 6 August 2020 in the state-regions conference), intends to strengthen a vision that considers health as the result of the development of a harmonious and sustainable human being, nature, and the environment (One Health).

This vision, recognizing the interconnection between the health of people, animals, and ecosystems, promotes the application of a multidisciplinary approach to address risks that originate from the interface between the environment, animals, and ecosystems.

Accidents and occupational diseases fall within one of the six macro-objectives of the Plan, for which an evaluation system is envisaged for monitoring the essential levels of assistance (LEA), paying attention to social and environmental determinants.

In this framework, the value of health emerges as a strategic sustainability tool for places of life and work, and the importance of preventive functions in strengthening individual and corporate resilience is underlined.

The innovation element of the National Prevention Plan (PNP) 2020–2025, in Italy, is represented by its support for Health Promotion—thus making the development of empowerment and capacity-building strategies transversal to all the Macro Objectives, as recommended by the international literature and the WHO.

### 1.2. Management Approach to Prevention and Total Worker Health

The current framework of changes in the world of work requires the adoption of more integrated intervention models, as also suggested by the “Global plan of action-WHO” which calls for addressing all aspects of workers’ health through the Healthy Workplace Model, to which the National Institute for Occupational Safety and Health (NIOSH) also refers, which itself proposed the Total Worker Health (TWH) program.

This program is defined by a set of policies, programs and practices that integrates the prevention of health and safety risks in the workplace, with its promotion in favor of broader worker well-being.

In fact, it also brings together all aspects of work in integrated interventions that collectively address worker safety, health, and well-being. Traditional occupational health and safety protection programs have primarily concentrated on ensuring that work is safe and that workers are protected from the harm that arises from the work itself. TWH builds on this approach through the recognition that work is a social determinant of health. Job-related factors such as wages, work hours, workload, interactions with coworkers and supervisors and access to paid leave impact the well-being of workers, their families, and their communities. The long-term vision of the TWH program is to protect the health and safety of workers and to advance their well-being by creating safer and healthier work.

Research on the Total Worker Health approach provides a scientific evidence base that can help businesses and communities reduce the impact and cost of injuries and illness, thereby helping to control healthcare costs and disruption to family and community life. The TWH approach promotes research into patterns of work organization and emerging forms of employment, recognizing that both occupational and non-occupational exposures can act together to produce worker illness and injury. By integrating the traditional focus on work-specific factors with attention to health conditions and the quality of working life, the TWH approach provides a pathway to improve worker creativity, innovation, and productivity by creating work and work environments that are safe, health-enhancing, meaningful and fulfilling.

To achieve the objectives of the TWH, it is a priority to increase awareness of “feeling good at work” and to promote dedicated educational interventions. In essence, it is necessary to plan integrated interventions aimed at achieving safe and healthy working conditions, with the involvement of all company figures of the prevention system: employers, managers of the Prevention and Protection Service (RSPP), occupational physician, Workers’ Representative for Safety (RLS) and the workers themselves.

In this context, on the corporate front, workplace health and safety management systems (SGSL) prove to be particularly effective tools for managing processes. The introduction of the SGSL is mainly due to cultural changes that have affected legislation on the topic, in which—over time—we have moved from a ‘Command and control’-type approach to a more managerial/organizational approach.

The origin of SGSLs lies in quality management systems as well as environmental management systems. A quality management system is the set of all connected and interdependent activities that influence the quality of a product or service. Likewise, the adoption of an SGSL guarantees that the protection of health and safety becomes an integral part of the management of a company, integrating health and safety objectives and policies into the design and management of work and production systems. It is based on an organizational culture that looks at health and safety not only as regulatory compliance, but as an integral part of work processes and a strategic lever for the overall improvement of company performance.

The foundation of the structure of an SGSL is the systemic vision in which an approach is identified for processes that interact with each other and are integrated in the management organization. Through the SGSL, the policy, strategies, health and safety objectives and methods to achieve them are defined, as well as the processes and a method for monitoring and measuring the results.

In addition to the implicit advantages inherent to the adoption of an SGSL (reduction of accidents and related costs, risk mitigation, better monitoring of dangers, etc.), if a company demonstrates that it has effectively adopted and applied an SGSL, it is relieved from criminal liability in the event of a fatal or serious injury.

An SGSL, as a tool for optimizing resources and preventing and managing risks, can represent a guide capable of promoting an organizational context that best enhances professionalism, promoting a participatory spirit and therefore considering the general organizational context.

The effectiveness of the SGSL is conditioned by its ability to permeate all levels of the organization, allowing all subjects involved to achieve and maintain high standards of care, involving organizational structures and processes, risk assessment and monitoring mechanisms, performance and quality assistance, continuous training, and professional evaluation.

### 1.3. Management Approach and ISO 45001 Standard

In this context, the ISO 45001: 2018 standard is rapidly spreading—the first certifiable international standard on workplace health and safety management systems.

The standard directs companies to broaden the perspective of risk management through assessments related to analysis of the context (and the interactions between the external and internal environment), the process of evaluating risk as an opportunity and the involvement of all interested parties, including workers and stakeholders.

The importance of active participation is also highlighted, also in relation to the reporting of near accidents. The objective of the company organization is to develop awareness of its role; the UNI ISO 45001 standard refers to this when it “talks about worker leadership”.

A sense of responsibility is cultivated with good training, active participation, and constant positive feedback from the company towards its workers. To this end, communication and information processes are very important for workers and citizens, but also for all decision-making and managerial figures involved in risk management, constituting the fundamental element of the management approach that is represented by the participation of all parties involved.

We are witnessing the creation of a new cultural path for framing the risks present in companies within a broader framework that considers environmental, social, and economic factors in approaches for managing them, placing the human factor at the center of policies and prevention actions in an organic vision that supports institutions and companies in balancing these risks.

In this context, the final objective of this review is to highlight the opportunities of and critical issues in the application of management approaches for the prevention of accidents and occupational diseases, also in relation to the possibility of the continuous improvement of health and safety conditions in workplaces. Importance was also given to the evaluation of the effectiveness of management approaches and the description of related indicators.

## 2. Materials and Methods

A literature review was conducted to extract information on “management approach” and “management systems and organizational models for health and safety in the workplace”.

An electronic bibliographic search using the National Center for Biotechnology Information (NCBI) Databases (PubMed and MEDLINE (Ovid) was performed in January 2023.

Research topics concerned management approaches and management system, with different elements related to them (for example, worker participation, leadership, involvement of social partners).

An additional hand search of some articles was made for older references using some features of the themes All articles chosen were published in English.

Another research subject focused on the evaluation of the effectiveness of prevention interventions and related impact indicators.

In addition to the literature review, a review of our articles and experiences was also presented to discuss the features of management approaches, with reference to targeted prevention plans.

The analysis of the works presented in this review is not necessarily exhaustive, reporting on some of the opportunities and criticalities of management approaches, but it attempts to provide a comprehensive general picture of the various factors involved in a very complex topic such as that of the application of management approaches to health and safety in the workplace.

## 3. Relevant Sections

The UNI ISO 45001 standard introduces, in continuity with what has already been mentioned in the national social legislation, an approach that guarantees the efficiency and continuous improvement of management systems to respond to the evolution of the context in which a business organization operates.

In the field of health and safety at work, only in recent years has it been possible to consider the external factors that can influence companies’ choices and the proposal of interpretative models [1,2,3].

The growing interest of researchers and OSH prevention operators in the areas in which companies carry out their production or service activities is directly linked to methodological insights. In fact, as observed by various authors [4,5], context analysis represents an important phase not only for the correct planning and implementation of interventions, but also for understanding the reasons for their preventive effectiveness.

The Context–Mechanisms–Outcome (CMO) model, applicable for verifying the effectiveness of prevention interventions [6,7], defines the context as the set of characteristics and conditions of criticality or opportunity in which a specific intervention is carried out to achieve set objectives.

Other authors [8] highlight how the analysis of context can provide useful elements for identifying the factors that favor the drive for improvement in the companies involved. Among the mechanisms considered by companies to be the most effective for stimulating changes at work are legislative compliance, economic incentives and information/training on technical standards and guidelines and good procedural, technical and organizational practices. These considerations allow for a reflection on the models that can be adopted for the implementation of prevention interventions. Among the models proposed in the literature, the Driving Force–Pressure–State–Exposure–Effect–Action (DPSEEA) model proposed by Hambling et al. [9] has been used in Italy for the analysis of the effectiveness of accident-prevention plans in the construction sector [10].

The actions in the model fall within, among others, the following strategic objectives outlined in Italy in the current National Prevention Plan 2020–2025:The development of surveillance systems for accidents and occupational diseases.Strengthening and improving the coordination and standardization of supervisory activities.The promotion of health surveillance.The strengthening of communication and assistance processes for companies.The development of relationships with bilateral bodies as well as social and professional partners.The dissemination of good practices and promotional and training activities in schools.

Within this framework, and the indications that emerged from the analysis of the international literature, the opportunity was identified of stimulating motivation for OSH management improvement in companies by encouraging the active participation of the Institutions–Social Partners network in the implementation of targeted improvement plans, to assist companies in the process of improving the assessment and management of accident and health risk factors.

Knowledge of the context and the expectations of the interested parties allows for the definition (or modification) of strategies for the development and consolidation of health and safety, as also proposed by Masi et al. [11] and Schwartz et al. [12].

Another new aspect is given by the emphasis placed on the ‘Leadership and commitment’ of the company’s top management [13]. Top management must establish the OSH policy, objectives, and directions, as well as the necessary resources.

The assistance of institutions and social partners can help support company leadership in overcoming, or at least reducing, the various critical issues that hinder the application and successful outcome of prevention interventions, as discussed by Masi et al. [14] and Hasle et al. [15].

The theme of coordination and integration between institutions and social partners in the implementation of prevention assistance programs has stimulated, in recent years, a growing interest in the scientific debate on the need to verify the effectiveness of assistance interventions through the application of experimental protocols. Such protocols should consider, on the one hand, the necessary scientific rigor, and on the other, the business contexts in which interventions are carried out, which entail critical issues in the standardization of protocols—especially for small-sized companies. 

A methodological reference to the evaluation of management effectiveness, previously developed for organizations in the healthcare sector, is represented by the Donabedian model [16,17]—a still widely recognized conceptual scheme for examining healthcare services and evaluating the quality of care using performance indicators.

The model is represented by the paradigm of the three structure–process–result elements, which represent the three macro indicators used to draw conclusions on the quality of care in each system. The framework includes and measures the quality of the structural, technological, organizational, and professional factors that influence the context in which care is provided. The process measures, based on reference standards, the appropriateness of the care process as the sum of its various components. Process indicators (leading indicators), if correlated with recommendations based on robust evidence, allow us to predict improvements in or the worsening of health interventions (prevention, diagnosis, therapy, rehabilitation, assistance). The outcome, measured through lagging indicators, contains all the effects of health interventions on patients or populations, including changes in health status, behavior, or knowledge, as well as patient satisfaction and quality of life related to health.

The use of indicators for measuring the performance of all phases of an intervention is referred to by numerous authors as a tool for recording qualitative and quantitative data in the planning, monitoring, training, and impact stages, in terms of the improved solutions developed.

Building local networks by linking different organizations together is essential to achieving sustainable OSH improvements [18]. To this end, a fundamental element will be the involvement of homogeneous groups of companies (supply chain, cluster, district, etc.) through collaboration with local organizations and associations representing employers and workers.

The effectiveness of a management approach is, without a doubt, conditioned by its ability to permeate all levels of the organization—to allow all those involved to achieve and maintain high standards of care.

The PNP 2020–2025, in the macro-objective 5.4 ‘Injuries and accidents at work, professional diseases’, identifies the territorial intervention model (Targeted Prevention Plan—PMP), as an operational tool for organizing support actions for the evaluation process of risks and the organization of prevention and protection activities to improve company OSH performance.

The national and regional policy documents (guideline, vademecum, checklist) drawn up, starting from a specific need for protection, can allow us to extrapolate specific organizational, technological, and structural solutions that are concretely implementable and transferable, for the improvement of the health and safety of workers in a company. 

Taking into account a combined evaluation of evidence such as trends in the number of accidents (including fatalities), trends in reports of occupational diseases, evidence of non-compliance and the regional socio-economic aspects of production, it is possible for each Region to identify some specific ‘areas’ in which to intervene (with a PMP), either according to a proactive approach by the Local Health Authority Services responsible for protecting the health and safety of workers or oriented towards assistance to businesses.

The Targeted Prevention Plan represents an innovative control tool based on the conduct of prevention processes aimed at improving general protection measures, and not just at verifying the application of the rule. The action of Services for the protection of the health and safety of local health workers is oriented, in fact, towards support/assistance in the world of work, facilitating companies’ access to knowledge—or rather to the evaluation and correct management of risks—to reach, above all, small and medium-sized enterprises, which make up a large part of the Italian productive fabric.

One of the most debated topics among prevention operators concerns the ability to propose effective assistance models aimed at SMEs. A recent report promoted by EU-OSHA (SESAME Project-EUOSHA, 2018) indicates the need to develop more experimental experiences in the application of assistance models in companies that include the verification of the effectiveness of the results achieved in terms of improvements in company performance, with special attention paid to the analysis of near misses as sentinel indicators of critical issues in a company.

National experiences relating to the implementation of targeted prevention plans have made it possible to evaluate the effectiveness, on an organizational side, of the synergies between institutions and companies and, on the technical side, in improving risk analysis, evaluation and management capabilities [19,20]. It is therefore essential to strengthen the support action for SMEs to improve decision-making and application capabilities for preventive actions, with particular attention paid to the process of reporting and analyzing near misses.

The improvement of regulatory applicability in small businesses can in fact be aided by more than the simplification of legal documentary obligations, stimulating the motivation to improve health and safety levels and the skills of employers and company RSPPs by making self-analysis methodologies and tools—which provide the Employer with data and information useful for undertaking preventive actions—that are also based on a cost–benefit perspective, for use in company management choices.

In applying a management approach, in fact, the organization will have to provide the technical, economic, and human resources necessary to ensure that the SGSL is established, implemented, maintained, and continuously improved in a dynamic cycle. It is therefore essential that it determines the skills of workers and ensures that they are able, based on the level of education and specific training acquired, to identify dangerous situations.

In such an articulated context, it is essential to have tools that can be inserted naturally within one or more phases of an SGSL. Tools which, beyond specific needs, must allow for a methodology for the assessment and management of risks, allowing critical issues to be quickly identified and appropriate corrective measures to be developed.

In the management approach indicated by UNI ISO 45001: 2018, the critical issues found in process audits, integrated with analyses of data relating to the causal risk factors of near misses and accidents (lagging indicators), allow for the application of improvement programs to verify process indicators (leading indicators) for use in planning objectives.

For the purposes of this document, process indicators are proactive measures that provide information on the effective performance of health and safety activities. They measure activities that may contribute to injuries, illnesses and other incidents and reveal potential problems in a health and safety program. In contrast, outcome indicators measure the occurrence and frequency of events that have already occurred, such as the number or rate of injuries, illnesses, and deaths. While outcome indicators can alert you to a critical issue in an area of your health and safety program, or the existence of a hazard, process indicators allow you to take preventative action to address that critical element before it turns into an incident. A good management program uses process indicators to direct improvement actions and outcome indicators to measure their effectiveness. The evaluation of effectiveness through this monitoring process and performance measurements determines whether an institutional support intervention for health and safety has achieved the set objectives. The applied approach is oriented towards a pre–post-evaluation of effectiveness and is designed with the aim of collecting—through a set of qualitative–quantitative indicators—the data and information detected during the carrying out of the various phases of the individual PMP, and through result indicators, the improvement actions implemented by the support system network and participating companies [18].

The detailed information that emerges from the detection of individual risk factors (pre- and post-event) aims to detect both the proximate causes (for example, problems with equipment, incorrect operating methods, inadequately prepared work environments, etc.) and root causes of the events to activate the consequent flows in the company’s processes for the management and containment of the risks that have emerged.

Starting from the knowledge provided by the Informo national surveillance systems and Mal Prof, developed in collaboration with Regions and Inail Dimeila, various tools have been created for the use of companies for the in-depth analysis of risk factors present in the workplace and for the management of reporting procedures and the analysis of near misses [21], in order to assist companies in managing health and safety and to identify effective models for the participatory prevention interventions implemented by institutions and social partners.

## 4. Discussion

In applying a management approach, organizations will have to provide the technical, economic, and human resources necessary to ensure that the SGSL is established, implemented, maintained, and continuously improved in a dynamic cycle. It is therefore essential that it determines the skills of workers, based on the level of education and specific training acquired, to identify dangerous situations.

In this context, it is an advantage to have specific tools that can be easily inserted into SGSL phases—tools which, beyond specific needs, will allow us to have a methodology for assessing and managing risks, allowing us to quickly identify critical issues and develop appropriate corrective measures [22].

The most effective approach for dealing with risks in working environments is the management approach, explained at different levels in the studies highlighted in this review, which integrates and systematizes all the elements that contribute not only to the health and safety of workers, but also to well-being of the person at the center of the work activity.

The concepts introduced by the “Donabedian model”, in line with the management approach to health and safety in the workplace and the broad vision referred to the corporate Total Work Health mentioned above, can be related to the concept of a “syndemic”, proposed by medical anthropologist Singer and understood by the author as a new approach to prevention policies based on the consideration of the elements of social and economic contexts that can contribute to health emergencies related to the employment context [23].

The interactions between accidents, occupational pathologies and economic, social, and environmental factors can, in fact, also be described through a “syndemic approach”, which examines the health consequences of the interactions in specific contexts.

Understanding the mechanisms of mutual influence of these factors is important for prevention, solutions, and policies, and requires an effort towards developing a general awareness of diseases and injuries, their groupings, and their synergies in biological, ecological, and social contexts in terms of broad-based public health policy initiatives with prevention objectives.

To achieve this ambitious objective, an approach to health and safety is necessary that goes beyond the mere regulatory prescription and that contemplates the organizational and technical context in which each working reality is immersed, in a holistic approach that integrates the different aspects of the health and well-being of everyone.

For this purpose, the commitment of the institutions and social partners involved is necessary to support companies in the process of the continuous improvement of health and safety conditions in the workplace, in which the various entities involved collaborate in a network through reciprocal exchange [24].

Based on the approach described above, in recent years, new paradigms for the integration of health promotion, the prevention of work-related disease and the organization of work have been developed in many countries.

Integrated interventions are shown to increase overall health and safety at worksites more effectively and rapidly than more narrowly focused programs [25], underlining the importance of planning phases in addition to implementation and evaluation. The themes highlighted have included organizational priorities, leadership buy-in, external pressures, training, program promotion and evaluation metrics [26].

Finally, another important element for the proper functioning of management approaches is represented by the analysis of the context in which the organizational interventions are applied [27].

## 5. Conclusions and Future Directions

Given the qualifying elements of management approaches analyzed in this review, some critical elements remain, which at the same time, also represent an opportunity for in-depth analysis and greater knowledge.

As regards the institutional level, for example, some fundamental questions remain open: more effective coordination of the bodies involved to avoid an overlap of roles, functions and activities; the integration of information into complex systems and, therefore, the consolidation of surveillance systems as a source of information on accidents and occupational diseases; the development of tools and methods for the assessment and management of risks tailored to specific company situations and the planning of interventions in relation to the subsequent evaluation of effectiveness through monitoring campaigns.

At the company level, some focal points can be further evolved, such as the greater sensitivity of the leadership in defining the most appropriate organizational choices; the clear awareness of workers regarding risks and measures to combat them and the real participation of all company subjects, including in the reporting of near misses, for the purpose of more effective and timely prevention.

The hope is to see ever-greater integration between the institutional and corporate levels to aspire to continuous improvement in health and safety conditions and to aim for the complete well-being of workers.

To plan prevention policies and interventions, the recent TWH approach allows us to adequately consider the synergy between risks related to work, the environment, lifestyles, and personal health conditions.

The purpose of the TWH is, in fact, to pursue a working environment free from risks and dangers that could compromise the safety of workers, and, at the same time, to develop policies, programs and practices for the total well-being of workers [28].

## Data Availability

Not applicable.

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
