# Peer review of "Management Approaches to Health and Safety at Work during Prevention Intervention Planning"

_ijerph, 2023, doi:10.3390/ijerph20247142_

Round 1

Reviewer 1 Report

Comments and Suggestions for Authors

I appreciate that the authors compiled a considerable amount of information for the article.  However, there are several issues.  The manuscript presents this work as a "systematic literature review" but a major ommission is that there is no description of the methodology that was used to conduct it.  Also, there were some difficulties with the communication of the information.  very long sentences (with an excessive and unnecessary words) are used in many places throughout the manuscript, and the point that the authors may be trying to make are lost.  The writing needs to be clearer and more concise.  Also, decrees and other regulations are discussed, but not enough background for these are given, and the Total Worker Health framework is not adequately described. 

Comments on the Quality of English Language

There were several paragraphs that consisted of very long, confusing sentences, and some cases, the wrong words/terms may have been used. The manuscript For example, 43-54, 60-62, 66-71, 75-80 need to be edited for better clarity.

Author Response

A section relating to the methodology used for the review has been added.

Topics and paragraphs have been better divided and, in some parts, longer or more complicated sentences have been simplified

Total Worker Health framework  has been described in more detail

The changes and additions made are highlighted in yellow.

Reviewer 2 Report

Comments and Suggestions for Authors

Dear authors, thank you for the opportunity to familiarize yourself with the results of your analytical research.

Your article is dedicated to a current topic and certainly deserves attention.

The introduction demonstrates the relevance of the study and substantiates the importance of studying this problem.

At the same time, the introduction does not clearly indicate the purpose of the study. It should be added, as well as an explanation of what documents the authors worked with and how they were selected for the purposes of the analytical review.

In this connection, the chosen structure of the manuscript is unclear. Authors should structure the main part of the article into subsections.

The discussion of the results is very brief, including only one reference. Although the discussion of the results involves an analysis of the findings in relation to other studies. Needs to be expanded.

The limitations of the present study need to be described.

The list of sources is very small for a review article (only 21 sources). It requires expansion in terms of the analysis of studies demonstrating the effectiveness of certain approaches. To further substantiate the authors' conclusions.

In this connection, the article requires significant revision.

Best wishes, reviewer

Author Response

The purpose of the study was clearly stated in the introduction..

The article has been better divided into subsections.

The discussion of the results and its references have been expanded.

The limitations of the study have been described in the Material and Methods paragraph.

The list of source has been expanded.

All changes and additions made are highlighted in yellow.

Best regards

Round 2

Reviewer 1 Report

Comments and Suggestions for Authors

The manuscript is much improved with the additions.

Comments on the Quality of English Language

There are still fairly long sentences contained in the manuscript.  Minor editing is  recommended.

Author Response

We have semplified some sentences as in revised version in attach.

Best regards

Reviewer 2 Report

Comments and Suggestions for Authors

Dear authors, thank you for your corrections and additions.

The article may be recommended for publication

Best regards, reviewer

Author Response

We are semplified some sentences as requested by reviewer 1.

Best regards
